# Evaluating the Effects of Diet-Gut Microbiota Interactions on Sleep Traits Using the UK Biobank Cohort

**DOI:** 10.3390/nu14061134

**Published:** 2022-03-08

**Authors:** Xin Qi, Jing Ye, Yan Wen, Li Liu, Bolun Cheng, Shiqiang Cheng, Yao Yao, Feng Zhang

**Affiliations:** 1Precision Medicine Center, The First Affiliated Hospital of Xi’an Jiaotong University, Xi’an 710061, China; xinqi0702@xjtufh.edu.cn; 2Key Laboratory of Trace Elements and Endemic Diseases of National Health and Family Planning Commission, School of Public Health, Health Science Center, Xi’an Jiaotong University, Xi’an 710061, China; 1301550385@163.com (J.Y.); wenyan@mail.xjtu.edu.cn (Y.W.); liuli0624@stu.xjtu.edu.cn (L.L.); cblbs1@stu.xjtu.edu.cn (B.C.); chengsq0701@stu.xjtu.edu.cn (S.C.); yao3077690800@stu.xjtu.edu.cn (Y.Y.)

**Keywords:** dietary compositions-gut microbiota interactions, dietary habits-gut microbiota interactions, polygenetic risk score, insomnia, sleep duration

## Abstract

Previous studies showed that diet and gut microbiota had a correlation with sleep. However, the potential interaction effects of diet and gut microbiota on sleep are still unclear. The phenotypic data of insomnia (including 374,505 subjects) and sleep duration (including 372,805 subjects) were obtained from the UK Biobank cohort. The Single Nucleotide Polymorphisms (SNPs) associated with 114 gut microbiota, 84 dietary habits, and 4 dietary compositions were derived from the published Genome-wide Association Study (GWAS). We used Linkage Disequilibrium Score Regression (LDSC) to estimate the genetic correlation and colocalization analysis to assess whether dietary habits and insomnia/sleep duration shared a causal variant in a region of the genome. Using UK Biobank genotype data, the polygenetic risk score of gut microbiota, dietary habits, and dietary compositions were calculated for each subject. Logistic regression and linear regression models were used to assess the potential effects of diet-gut microbiota interactions on sleep phenotypes, including insomnia and sleep duration. Insomnia and sleep duration were used as dependent variables, and sex, age, the Townsend Deprivation Index scores, and smoking and drinking habits were selected as covariates in the regression analysis. All statistical analyses were conducted using R-3.5.1 software. Significant genetic correlations were discovered between insomnia/sleep duration and dietary habits. Further, we found several significant dietary compositions-gut microbiota interactions associated with sleep, such as fat × G_Collinsella_RNT (*p* = 1.843 × 10^−^^2^) and protein × G_Collinsella_HB (*p* = 7.11 × 10^−^^3^). Besides, multiple dietary habits-gut microbiota interactions were identified for sleep, such as overall beef intake × G_Desulfovibrio_RNT (*p* = 3.26 × 10^−4^), cups of coffee per day × G_Escherichia_Shigella_RNT (*p* = 1.14 × 10^−3^), and pieces of dried fruit per day × G_Bifidobacterium_RNT (*p* = 5.80 × 10^−3^). This study reported multiple diet-gut microbiota interactions associated with sleep, which may provide insights into the biological mechanisms of diet and gut microbiota affecting sleep.

## 1. Introduction

Insomnia, one of the common types of sleep disorders, is characterized by difficulty initiating or maintaining sleep, or non-restorative sleep, which is a symptom of dissatisfaction with the quality or duration of sleep [1,2]. Insomnia may be caused by many factors, such as stress, medical conditions, the use of certain drugs, poor sleep habits, eating too much in the evening, or the change of schedule due to work or travel [1]. Insomnia affects approximately 20% of adolescent and 35% adults [3,4,5]. It can cause fatigue, concentration or memory impairment, and motivation or energy reduction [1]. Sex differences exist across insomnia and the number of women who suffer from insomnia is 1.5 times greater than the number of men [6,7]. Due to its high prevalence and its subsequent impairments, insomnia is an important public health problem, and it increases the risks of anxiety disorders, major depression, cardiovascular disease, and coronary heart disease [8,9,10]. 

Genetic, environmental, behavioral, and physiological factors are involved in the etiology and pathophysiology of insomnia [1,11]. For instance, the increased level of cortisol caused by insomnia could enhance the activity of the hypothalamic-pituitary-adrenal (HPA) axis. Compared with the general population, insomnia patients had a higher expression level of the cortisol awakening response (CAR) [12]. Moreover, the metabolic rate, body temperature, and autonomic nervous system response were all increased in insomnia patients [13,14,15]. Genetic factors have a largely stable influence on insomnia symptoms for adults over time, and the estimated heritability of insomnia is approximately 38% in males and 59% in females [16]. Family and twin studies demonstrated that insomnia had a significant familial aggregation with a high heritability, despite the exclusion of psychiatric disorders [17,18]. 

It has long been appreciated that the gut–brain axis plays an important role in maintaining homeostasis, and gut microbiota is one of the key regulators of gut–brain axis function [19,20,21]. The gut is the largest residence of human microbiota, with a unique combination for every individual [22]. Genetic factors can influence the development of gut microbiota, which has a profound effect on the sleep and mental states of the host via the microbiota–gut–brain axis [23]. In addition, it is demonstrated that the composition, diversity, and metabolic function of gut microbiota show a significant difference between insomnia patients and healthy controls, and *Bacteroides* and *Clostridiales* are regarded as the most important biomarkers to distinguish insomnia patients from healthy individuals [24]. More interestingly, the previous study used deep sequencing to analyze circadian rhythms in the fecal microbiota of mice and found that an abundance of *Bacteroidetes* showed circadian rhythmicity, which was more pronounced in female mice [25]. Numerous studies have suggested that gut microbiota is implicated in the development of insomnia, but the exact mechanism is not yet clear.

Previous studies have found that the quality and style of diet are associated with mental disorders and the high prevalence of insomnia [26,27]. Researchers discovered that a vegetarian diet is a risk factor for mental disorders [28]. Additionally, a ketogenic diet has profound effects on multiple targets implicated in the pathophysiology of mood disorders [29]. Evidence shows that improving diet quality may have an effect on gut health, which also benefits mental health [26]. In addition, Cecilia et al. found that a Mediterranean-style diet was associated with adequate sleep duration and fewer insomnia symptoms [30]. However, the biological mechanism of dietary habits affecting the risk of insomnia remains elusive for now. Furthermore, it is well known that dietary habits could significantly affect gut microbiota [31]. Therefore, it is reasonable to infer that dietary habits contribute to the development of insomnia through gut microbiota. 

In this study, we aimed to investigate whether the diet-gut microbiota interactions had an influence on insomnia and sleep duration and to identify the specific diet-gut microbiota interactions on sleep. Therefore, we performed polygenetic risk score and logistic/linear regression models for the individuals derived from the UK Biobank. Our findings may help understand the contributions of diet-gut microbiota interactions to sleep and provide novel insights for further research.

## 2. Methods

### 2.1. Insomnia and Sleep Duration Phenotypes in the UK Biobank 

#### 2.1.1. Ethics Approval

The UK Biobank study was approved by the National Health Service National Research Ethics Service (11/NW/0382). All of participants in the UK Biobank study have supplied and signed the informed consent forms. Our data dictionary ID for application is 46478.

#### 2.1.2. Study Population and Phenotype Definition

A cohort study of ~500,000 individuals from the United Kingdom, age 40 to 69 years, was obtained from the UK Biobank source. From over 9.2 million invitations, 503,325 participants were recruited. The extensive phenotypic data upon baseline assessment were reported by participants using touchscreen tests and questionnaires, as well as nurse-led interviews. A total of 372,805 individuals were included for the sleep duration phenotype (200,075 females and 172,730 males, 56.97 ± 7.94 years), while 106,773 cases and 267,732 controls (56.98 ± 7.93 years) were included for the insomnia phenotype in our analysis.

Insomnia and sleep duration were two self-reported phenotypes for which participants answered questions using a touch-screen computer. For insomnia, the question was, “Do you have trouble falling asleep at night or do you wake up in the middle of the night?” with the following response options: “Never/Rarely,” “Usually,” “Sometimes,” and “Prefer not to answer.” The “Never/Rarely” and “Sometimes” responders were labeled as “no insomnia symptoms.” “Usually” responders were taken as “with insomnia symptoms”. “Prefer not to answer” responders were excluded in our analysis. For sleep duration, the question was, “About how many hours’ sleep do you get in every 24 h?” with responses listed in hourly increments.

#### 2.1.3. Genotype, Imputation, and Quality Control

Genotyping were conducted using the UK BiLEVE Axiom Array or the UK Biobank Axiom Array [32,33]. PLINK 2.0 and R-3.5.1 software were used to perform quality control [34]. We removed the participants whose genetic gender was different from self-reported gender, or who were genotyped but not imputed, or who withdrew their consent. Additionally, individuals were restricted to only ‘white British’ based on self-reported ethnicity (UK Biobank field ID: 21000). We also used an estimator implemented using KING software to screen out genetically related individuals [35]. A total of 450,580 samples (244,945 females and 205,635 males) passed the sample quality control. Using the 1000 Genomes Phase 3 dataset as the reference panel, SHAPEIT3 software was used to perform imputation [36,37]. The imputation files were released in the BGEN (v1.2) file format [38]. The detailed description of the genotyping, quality control, and imputation were described in previous studies [39].

### 2.2. GWAS Dataset of Gut Microbiota

The SNPs associated with gut microbiome were derived from three independent GWAS of gut microbiota, including the Flemish Gut Flora Project (FGFP) cohort (*n* = 2223) and two German cohorts (Food-Chain Plus (FoCus, *n* = 950)), and the PopGen cohort (*n* = 717) [40,41,42]. For the FGFP cohort, the V4 region of the 16S rRNA gene was amplified and the Illumina HiSeq platform was used to perform sequencing with 500 cycles (HiSeq-Rapid SBS v2 sequencing kit), producing 2 × 250-bp and paired-end sequencing reads. LotuS pipeline and DADA2 pipeline (v.1.6) were both used to analyze the fastq sequences for every sample [43,44]. Two different arrays, the HumanCoreExome v1.0 and the HumanCoreExome v1.1, were used to genotype, and allele calling was performed using GenomeStudio v2.0.4. After quality control, a total of 509,886 variants remained. After phasing by SHAPEIT3, IMPUTE4 was used to impute using the UK10K and 1000 Genome Project Phase 3 samples as the reference panel [36,45,46]. The FoCus and PopGen cohorts were genotyped using the Illumina Omni Express + Exome array and the Affymetrix Genome-Wide Human SNP Array 6.0, respectively [41,42]. Data processing was also performed using DADA2. A detailed description of the cohorts, genotyping, imputation, and quality control approaches can be found in the previous study [40]. For microbial traits, they included two phenotypes, a presence/absence (P/A) phenotype and a zero-truncated (all zero values set as missing) abundance (AB) phenotype. Rank-normal transformation (RNT) and hurdle binary (HB) are two microbial trait models for continuous (AB) and binary (P/A) traits. There were 1056 linkage disequilibrium (LD) independent lead SNPs significantly associated with microbial traits identified and used to calculate the PRSs of 114 gut microbiomes in this study.

### 2.3. GWAS Dataset of Dietary Compositions 

The GWAS of four dietary composition phenotypes were used in our analysis, including relative intake of fat, protein, carbohydrates, and sugar [47]. The sugar phenotype included 235,391 individuals, while the fat, protein, and carbohydrate phenotypes all included 268,922 individuals. All individuals are of European ancestry. The survey instruments were a food frequency questionnaire (FFQ) and 24 h dietary recall (24HDR). Genotyping was performed using Illumina or Affymetrix Axiom UKBiobank. IMPUTE2, IMPUTE4, BEAGLE and MaCH/Minimac were used to perform the imputation, with the reference of 1000 G and HRC 1.0. The *p* value of all significant SNPs were <5 × 10^−8^. 6, 7, 10, and 13 LD independent lead SNPs significantly associated with fat, protein, sugar, and carbohydrates. More detailed information about four dietary composition phenotypes can be found in the previous study [47]. 

### 2.4. GWAS Dataset of Dietary Habits 

A recent large-scale GWAS summary dataset of dietary habits was used here [48]. The genotyping, imputation, and initial quality control on the genetic dataset have been described previously. Cole et al. performed the GWAS analysis for the intake of 85 single foods, analyzed as single food intake quantitative traits (FI-QTs), and 85 principal component-derived dietary patterns (PC-DPs) using the Food Frequency Questionnaire (FFQ) data in up to 449,210 Europeans from the UK Biobank. For all variables, BOLT-LMM software (v2.3.2) was used to conduct linear mixed model association testing to account for relatedness [49,50]. A total of 814 LD independent loci (defined as >500 kb apart, *p* < 5.0 × 10^−8^) were significantly identified for 84 dietary habits, including 42 PC-DPs and 42 single FI-QTs.

### 2.5. Genetic Correlations with Dietary Habits and Co-Localization

We used Linkage Disequilibrium Score Regression (LDSC, https://github.com/bulik/ldsc, accessed on 19 January 2022) to estimate the genetic correlation between insomnia/sleep duration and dietary habits, based on the GWAS summary statistics of insomnia/sleep duration and 84 dietary habits. LDSC is usually used to estimate heritability and genetic correlation from GWAS summary statistics. For the correlation of insomnia/sleep duration and dietary habits, we obtained published GWAS summary statistics for dietary habits (N = 455,146 individuals), insomnia (N = 386,533 individuals), and sleep duration (N = 446,118 individuals) in the UK biobank [48,51,52]. Dashti et al. identified 78 loci associated with sleep duration, and Jansen et al. found 202 loci associated with insomnia [51,52]. These GWAS summary data were derived from the website (http://sleepdisordergenetics.org/, https://ctg.cncr.nl/, http://www.kp4cd.org/dataset_downloads/t2d accessed on 19 January 2022). As recommended, the LD scores from individuals of European ancestry from 1000 G were calculated for SNPs in the HapMap 3 SNP set and used to calculate genetic correlation; *p* < 0.05 was selected as the significant level.

Then, for the significant dietary habits having genetic correlation with insomnia/sleep duration, colocalization analysis was used to assess whether the dietary habits and insomnia/sleep duration shared a causal variant in a region of the genome. Bayesian testing was used to perform colocalization analysis [53]. We used an approximation based on the minor allele frequency (MAF) of the SNP and ignored any uncertainty imputation, due to only *p* value available for our GWAS summary datasets and without regression coefficients and variances. The approximate Bayes Factor was used to compute for colocalization based on the dietary habits and insomnia/sleep duration GWAS summary datasets [54]. Five hypotheses were included for each possible pair of vectors (for traits 1 and 2): H0 (no causal SNP), H1 (only association with trait 1), H2 (only association with trait 2), H3 (two independent causal SNPs), and H4 (one shared causal SNP). As recommended, we defined the variants as colocalized when the posterior probability of a colocalized signal (SNP.PP.H4) was >0.1 [54]. Bayesian co-localization analyses were conducted by using the coloc R package [54].

### 2.6. PRS Analysis

In this study, to study the relationship between diet-gut microbiota interactions on insomnia and sleep duration, we calculate the PRS for each individual with insomnia and sleep duration groups based on the risk alleles associated with 114 gut microbiota, 84 dietary habits, and 4 dietary compositions, respectively. First, we defined *PRS_j_* as the PRS value of 114 gut microbiota for the *j* (*j* = 1, 2, 3, …, *k*) individual, defined as $PRS_{j}=\sum_{j=1}^{t}\beta_{m}SNP_{mj}$. $\beta_{m}$ represents the effect value of the risk allele of the *n*th (*m* = 1, 2, 3, …, *t*) significant SNP associated with 114 gut microbiota, which was obtained from previous research [40]. $SNP_{mj}$ denotes the risk allele dosage of the *m*th SNP of the *j*th individual. Then, the PRS of the gut microbiota was obtained for each individual in the insomnia and sleep duration groups, respectively. PRS analysis was performed using PLINK 2.0 [34]. PRS of 84 dietary habits and 4 dietary compositions were calculated with the same way. The PRS of diet-gut microbiota was defined as gut microbiota PRS × dietary habits/dietary compositions PRS. All the PRSs were standardized to have a mean of 0 and a variance of 1 prior to further analysis.

### 2.7. Statistic Analysis

We used the logistic regression model to analyze the associations between insomnia and diet-gut microbiota interactions. The insomnia phenotype was defined as a dependent variable, while dietary habits PRS, dietary compositions PRS, gut microbiota PRS, and diet-gut microbiota interaction PRS were respectively selected as independent variables. Sex, age, the Townsend Deprivation Index scores, and smoking and drinking habits were selected as covariates.

Additionally, the linear regression model was used to analyze the associations between sleep duration and diet-gut microbiota interactions in the same way. Sleep duration was selected as a dependent variable. All of these statistical analyses were conducted using R- 3.5.1; *p* < 0.05 was selected as the significant level.

## 3. Results

### 3.1. The Results of LDSC and Colocalization

We discovered multiple genetic correlation signals between insomnia and 84 dietary habits with *p* value < 0.05 (Appendix A). Further, for each significant dietary habit associated with insomnia, we identified the specific meaningful SNPs using the co-localization analysis, such as rs11693221 for bread type: whole meal/whole grain vs. any other (r_g_ = −0.2970, P_LDSC_ = 1.25 × 10^−23^, PP.H4 = 0.8153) and rs429358 for pieces of dried fruit per day (rg = −0.2587, P_LDSC_ = 2.24 × 10^−18^, PP.H4 = 1.00) (Appendix A).

For the genetic correlation of sleep duration and 84 dietary habits, we found several significant signals with *p* value < 0.05 and the specific SNPs were identified by colocalization analysis, such as rs62158206 for milk type: soy milk vs. never (r_g_ = 0.1461, P_LDSC_ = 4.32 × 10^−5^, PP.H4 = 0.2686) and rs6737318 for overall lamb/mutton intake (r_g_ = 0.0848, P_LDSC_ = 7.00 × 10^−4^, PP.H4 = 0.1319) (Appendix A).

Interestingly, 20 common genetic correlation signals shared by insomnia and sleep duration with dietary habits, such as bread type: white vs. any other (r_g_ = 0.3127, SE = 0.0251, *p* = 1.13 × 10^−35^ for insomnia; r_g_ = −0.0809, SE = 0.0261, *p* = 1.90 × 10^−3^ for sleep duration) and bowls of cereal per week (r_g_ = −0.2515, SE = 0.0267, *p* = 4.11 × 10^−21^ for insomnia; r_g_ = 0.0906, SE = 0.0231, *p* = 8.60 × 10^−5^ for sleep duration) (Figure 1).

### 3.2. The Results of Dietary Compositions-Gut Microbiota Interactions

For insomnia, we discovered several candidate dietary compositions-gut microbiota interactions, such as carbohydrate × G_Dialister_RNT (*p* = 1.78 × 10^−^^3^), fat × G_Collinsella_RNT (*p* = 1.84 × 10^−^^2^), and protein × G_unclassified_P_Firmicutes_RNT (*p* = 2.26 × 10^−^^2^) (Appendix A).

In addition, multiple dietary compositions-gut microbiota interactions were found for sleep duration, such as carbohydrate × O_Rhodospirillales_HB (*p* = 8.54 × 10^−^^4^), protein × G_Clostridium_sensu_stricto_RNT (*p* = 6.77 × 10^−^^3^), and protein × G_Collinsella_HB (*p* = 7.71 × 10^−^^3^) (Appendix A).

### 3.3. The Results of Dietary Habits-Gut Microbiota Interactions

We found multiple significant signals between insomnia and dietary habits-gut microbiota interactions, such as milk type: soy milk vs. never× G_Alistipes_RNT (*p* = 2.49 × 10^−4^), coffee type: decaffeinated vs. any other × G_Senegalimassilia_HB (*p* = 5.26 × 10^−4^) and tablespoons of cooked vegetables per day × G_unclassified_P_Firmicutes_HB (*p* = 3.24 × 10^−^^2^) (Appendix A).

For sleep duration, multiple significant interactions between gut microbiota and dietary habits were identified to be associated with sleep duration, such as bread type: white vs. any other × G_Clostridium_XVIII_RNT (*p* = 9.51 × 10^−^^5^), overall beef intake × G_Desulfovibrio_RNT (*p* = 3.26 × 10^−4^) and pieces of dried fruit per day × G_Bifidobacterium_RNT (*p* = 5.80 × 10^−^^3^) (Appendix A).

### 3.4. The Common Interactions of Sleep Duration and Insomnia

Interestingly, some common significant dietary habits/compositions-gut microbiota interactions were found both in sleep duration and insomnia, such as cups of coffee per day × G_Escherichia_Shigella_RNT (P_Sleep duration_ = 1.22 × 10^−2^, P_Insomnia_ = 1.14 × 10^−3^) (Appendix A). The top 20 common diet-gut microbiota interactions were showed in Figure 2.

## 4. Discussion

Although the effects of gut microbiota and diet on sleep have been explored in previous studies, the interaction effects of diet-gut microbiota on sleep remain unclear [23,27]. Therefore, we conducted logistic regression and linear regression models using the UK Biobank cohort to investigate the interaction effects of diet and gut microbiota on the development of insomnia and sleep duration issues. Interestingly, we identified multiple significant dietary compositions/habits-gut microbiota interactions related to sleep. Our results may provide novel clues for further research in the mechanism of insomnia and sleep duration.

We observed that dietary compositions/habits-gut microbiota interactions were associated with sleep. The composition of gut microbiota is a dynamic ecosystem shaped by dietary habits, and long-term dietary interaction can lead to the alterations of gut microbiota [55,56]. Gut microbiota has an influence on the brain through the microbiota–gut–brain axis, which has been shown to affect neuropsychiatric conditions [57]. Besides, there is growing evidence that gut microbiota is associated with neural development and function in the enteric and brain nervous system [58]. Gut microbiota can induce antigenic stimulation to shape the physiological immune response. Moreover, the expression levels or affinities of neuropeptides may be involved in neuropsychiatric conditions, including eating and sleep disorders [59]. A randomized controlled trial indicated that diet intervention could affect sleep quality through influencing the composition of gut microbiota [60]. 

Interestingly, in our study, we identified multiple dietary compositions-gut microbiota interactions associated with sleep, including fat × G_Collinsella_RNT and protein × G_Collinsella_HB. The effect size of fat × G_Collinsella_RNT and protein × G_Collinsella_HB were −0.009 and −0.004, which was minor. However, they were meaningful in the research mechanism for sleep. A controlled crossover study was performed in 10 healthy controls with average 2526 (±951) kcal/day of baseline total energy intake and consumption of a fat-rich fast food (FF, fat intake: 44.6 ± 4.0%) and high fiber Mediterranean diet (Med, fat intake: 33.2 ± 2.5%) in randomized order for 4 days each, with a 4-day washout between treatments. The results showed that Collinsella still showed a significant increase after consuming fat-rich food, even after the adjustment for multiple testing (fold change _(FF vs. Med)_ = 3.54, adjusted *p* = 0.028) [31]. Furthermore, after 19 overweight/obese participants were assigned to a low-carbohydrate, high-fat diet for 4 weeks, in which the protein and fat were 1.5 times higher compared to that in the normal weight reference group, the abundance of Collinsella decreased in parallel with a decrease in body mass index [61]. Then, compared to subjects with Bipolar Disorder-I, the abundance of Collinsella was regarded to be significantly different (*p* < 0.05 after Bonferroni correction) among Bipolar Disorder-II subjects using ANCOM (analysis of communities of microorganisms) [62]. In addition, the relative abundance of Collinsella was significantly increased in 82 schizophrenia subjects, compared with 80 normal controls (*p* = 0.001, FDR corrected) by 16S rRNA sequencing [63]. Collinsella was also found to have a difference (FC = 2.69, *p* = 0.004) in subjects with autism spectrum disorder compared with healthy controls [64]. Insomnia was observed frequently in the autism spectrum disorder, schizophrenia, and bipolar disorder subjects [65,66].

Dietary habits could also affect gut microbiota. We also found that there existed significant genetic correlations between diet and sleep. Multiple dietary habits × gut microbiota interactions were discovered to be associated with sleep, including the interactions between meat intake and gut microbiota. For example, the effect size of overall beef intake × G_Desulfovibrio_RNT and overall processed meat intake × G_Blautia_RNT was 0.006 and 0.002, and they had relationships with sleep. Alber et al. used a dataset of 1341 participants and found that each 100 g/d increment of meat intake was associated with a 60% higher risk for both large sleep duration changes and poor sleep quality (OR = 1.60; 95% CI = 1.07–2.40) [67]. Two randomized, controlled feeding studies manifested that when they used energy-restricted diets (a 750-kcal/d deficit), a higher proportion of energy consumption from protein (20% vs. 10%), such as beef, could improve the global sleep score (GSS) in overweight and obese adults, accessed by the Pittsburgh Sleep Quality Index (PSQI) questionnaire [68]. Meanwhile, the increase in Desulfovibrionaceae responds to low-fat diets (12% Kcal, 184.70 g/kg beef) as compared with high-fat diets (60% Kcal, 251.81 g/kg beef) in C57BL/6J mice [69]. Besides, through 16S rRNA sequencing, at the genus level, the level of Desulfovibrio was lower in the sleep disturbance group than that in the no sleep disturbance group (r = −0.448, *p* = 0.006) [70]. Compared with the baseline, the increment of red meat consumption was associated with Blautia (*p* = 0.039) [71]. Robert et al. found that Blautia was significantly correlated with sleep through the gut–microbiota–brain axis (r = −0.57, *p* < 0.05) in humans [72,73]. Combined with the previous studies, our findings suggested that interactions of overall beef intake × G_Desulfovibrio_RNT and overall processed meat intake × G_Blautia_RNT had relationships with sleep.

In addition, several interactions between fruit, vegetables, and gut microbiota were found to have correlations with sleep traits, including tablespoons of cooked vegetables per day × G_unclassified_P_Firmicutes_HB, tablespoons of raw vegetables per day × G_Prevotella_RNT and pieces of dried fruit per day × G_Bifidobacterium_RNT. The effect size of interactions for fruit and vegetables with gut microbiota was small due to less allelic variations at the associated SNPs in the application population. However, the research showed they are still worth study. The significant associations between vegetable-based diets and gut microbiota (Prevotella and Firmicutes) were detected in 153 subjects habitually following omnivore, vegetarian, or vegan diets [74]. Through a randomized within-subject crossover study, the researchers found that the sleep effect was driven by an almost doubled Firmicutes:Bacteroidetes ratio after two days of partial sleep deprivation (sleep opportunity 02:45–07:00 h) vs. normal sleep (sleep opportunity 22:30–07:00 h) (17.5 ± 13.7 vs. 9.1 ± 4.6, *p* = 0.04, one-sided) in nine normal-weight men [75]. A controlled clinical study showed that after 10 healthy women continuously consumed orange juice for 2 months, the composition and metabolic activity of microbiota altered and Bifidobacterium had an increase of 6.12 ± 0.84 to 6.87 ± 0.36 of log_10_ cfu·g^−1^ [76]. Moreover, compared with the control, Bifidobacterium decreased (fold difference = −0.9825, adjusted *p* = 0.005) in sleep-disrupted mice [77].

This study detected interactions between coffee and gut microbiota for sleep, including cups of coffee per day × G_Escherichia_Shigella_RNT. It has been demonstrated that gut microbiota can regulate neurotransmitter signals, and Escherichia species can produce noradrenaline [78]. Noradrenaline may affect rapid eye movement sleep (REMS) through modulating the transcription of many molecules, the process of which is complex and diverse [79]. Karacan et al. found that after 18 normal young adult males received 4-cup equivalents of regular coffee 30 min before bedtime, the coffee or caffeine may cause rapid eye movement sleep to shift to the early part of the night [80]. Both of coffee and Escherichia_Shigella had an indirect or direct relationship to rapid eye movement sleep.

Nevertheless, the evidence connecting diet-gut microbiota interactions and sleep is lacking. In this study, we used the UK Biobank data to explore the influence of diet-gut microbiota interactions on sleep duration and insomnia. Interestingly, we identified multiple significant diet-gut microbiota interactions associated with sleep. These findings indicated that modulating diet-gut microbiota interactions could provide a new sight for investigating the mechanisms of insomnia and sleep duration. However, there were several limitations that must be mentioned. First, because all individuals in this study were white, more attention should be paid by applying the significant results to other races. Second, as *p*-values depend upon both the magnitude of association and the sample size, “*p* < 0.05” is identified as a cut-off that indicates “statistical significance.” However, there are different views on how to correctly interpret the *p* value and examine the concept of statistical significance. Many conditions can result in the *p* values < 0.05, such as the large sample size. Studies indicated that the research based on small size would not provide precise enough estimates of the sample size needed to achieve the planned power [81]. Therefore, other and better ways would be suggested for analyzing data and for presenting, interpreting, and discussing the results. In some conditions, effect sizes are used to measure the degree of influence of a certain independent variable. In regression analysis, the effect size should be combined with *p* values, both of which depend on statistical significance [81]. The sample size used in this study is enough large (N = 372,805 for sleep duration; 374,505 for insomnia) and *p* values are small (all *p* < 0.05) in the results. We selected the *p* < 0.05 as the significant level. Moreover, in the present study, PRS has a minor effect on the target disease, the reason of which may be less allelic variations at the associated SNPs in the application population [82]. Therefore, the effect size of diet-gut microbiota interactions associated with sleep was minor. Our observational analysis was a preliminary screen for the significant interactions between diet and gut microbiota, and the findings need to be verified by further confirmatory experimental analysis.

## 5. Conclusions

In conclusion, using logistic regression and linear regression models, our results identified that multiple diet-gut microbiota interactions were associated with sleep. Additionally, our study provided insights for studying the mechanisms of insomnia and sleep duration issues. More detailed mechanisms revealing the links between diet-gut microbiota interactions and sleep should be further explored in the future.

## Figures and Tables

**Figure 1 nutrients-14-01134-f001:**
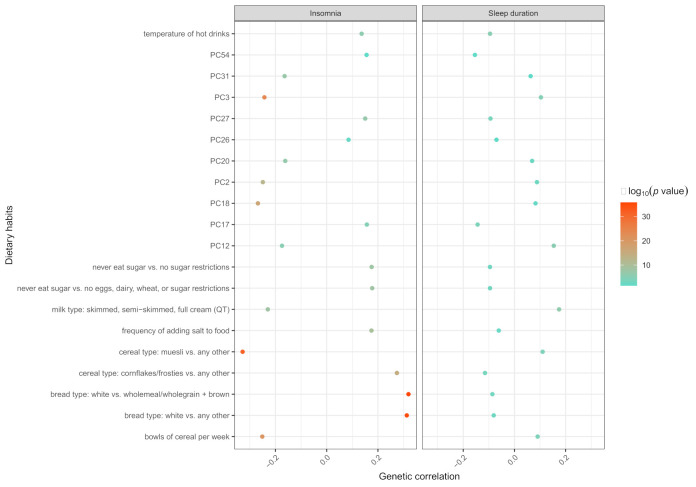
Twenty common genetic correlation signals shared by insomnia and sleep duration with dietary habits. The horizontal axis represents the genetic correlation (r_g_). Different colors indicate the −log_10_ (*p* value).

**Figure 2 nutrients-14-01134-f002:**
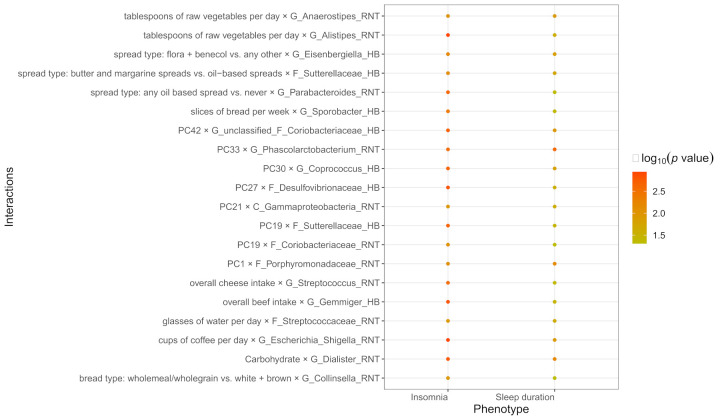
The top 20 common dietary habits/compositions-gut microbiota interactions shared by both insomnia and sleep duration.

## Data Availability

Availability of data and material data sharing is not applicable to this article, as no datasets were generated or analyzed during the current study.

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
