# Peer review of "Evaluating the Effects of Diet-Gut Microbiota Interactions on Sleep Traits Using the UK Biobank Cohort"

_nutrients, 2022, doi:10.3390/nu14061134_

Round 1

Reviewer 1 Report

The authors properly addressed the issues in the cover letter. The article has been also improved but still needs some modifications based on the authors' response letter. Please explain  in Discussion clearly about the limitations of p value in terms of sample size and refer the readers to Tables 1 and 2 that you presented in the response letter. These tables can be marked as the supplementary tables. If you like your article to be cited more frequently, it is important to make it more understandable in terms of the meaning of effect size by practical examples in Discussion. 

Reviewer 2 Report

I'd like to thank the authors for addressing all of my comments.

Author Response

We are grateful for giving us an opportunity to reconsider our revised manuscript entitled "Evaluating the effects of diet-gut microbiota interactions on sleep traits using UK Biobank cohort " (Manuscript Number: Nutrients-1468550) and take the publication in consideration. We appreciate all your time and effort in helping us improve and clarify our manuscript. 

This manuscript is a resubmission of an earlier submission. The following is a list of the peer review reports and author responses from that submission.

Round 1

Reviewer 1 Report

Overall this is a very well written manuscript. Below are my comments

Introduction

The introduction is well written however, in the final paragraph you should simply provide your objective and hypotheses and not the summary of what was done and what the results were.

Methodology

The methodology is very well written and detailed. I have a few questions for the authors which may help identify if there is a need for other analyses.

  1. Were the PRS that were calculated collinear? My previous work has found that there is significant collinearity between the variables that the authors presented. Please report whether the variables were collinear or not.
  2. If they was significant collinearity I would recommend the use of a cluster elastic net regression. LASSO might be another analysis, but literature suggests that cluster elastic net may be a better choice than LASSO when it comes to gut microbiota.

Results

Considering how many independent variables were used in this analysis the authors do an excellent job with presenting the results. I do question whether collinearity in the variables may have impacted the results. 

Discussion

Based on the results presented the discussion is sound however, I question whether the collinearity in the dependent variables may have influenced the findings. With that being said, if the results are maintained the content of the discussion was great however, there were significant grammatical errors (considering there were no line numbers in the manuscript I cannot provide exact line numbers). There were also formatting errors in terms of font (may not be the authors' fault as I have also experienced formatting issues when my Word document gets converted to Nutrients format). 

Reviewer 2 Report

This was an interesting article titled “Evaluating the effects of diet-gut microbiota interactions on sleep traits using UK Biobank cohort” and reported multiple diet-gut microbiota interactions associated with sleep. The research that is based on large sample size could over-relying on p-values to interpret findings. In this study of 500,000 observations, the p-values associated with the coefficients from modeling the data set are easily going to be 0. Then, the article should be cautious in assessing whether or not the small p-value is just an artifact of the large sample size, and carefully quantify the magnitude and the sensitivity of the effect. In other words, conclusions based on significance and sign alone, claiming that the null hypothesis is rejected, are meaningless unless interpreted in light of the actual magnitude of the effect size. The focus should be on the practical significance of findings rather than the sign and direction of regression coefficients. The article should be objective and clear in helping the readers understand the meaning of the coefficient estimates within the study context, i.e., the effect size. It should report the sensitivity of dependent variable to changes in independent variable, as a unit change in x is associated with an average change of β units in y in linear regression, and a unit change in x is associated with an average change in the odds of Y=1 by a factor of β in logistic regression. Also, because a large sample size results in tighter confidence interval, the thresholds of confidence interval are especially informative of the unknown parameter’s magnitude and range. In addition, the threshold p-value should be adjusted downwards as the sample size grows, however to the best of my knowledge there is no proposed rules-of-thumb in terms of how such adjustments should be made. Furthermore, by re-computing the p-value for a smaller sample, the article can show the possible changes in p values and the effect size.

Round 2

Reviewer 1 Report

I'd like to thank the authors for addressing my queries. I appreciate the authors trying to identify multicollinearity in their analyses. With such low R values, I think you're safe (especially with such a large sample). 

Before the manuscript is accepted, I'd like to see the authors add 95%CI to their tables so that future researchers may have an opportunity to interpret the analysis as well.

Author Response

We are grateful for giving us an opportunity to reconsider our revised manuscript entitled "Evaluating the effects of diet-gut microbiota interactions on sleep traits using UK Biobank cohort " (Manuscript Number: Nutrients-1468550) and take the publication in consideration. We have carefully addressed all the comments and detailed our responses in the following. We have also made related revision in the manuscript regarding to your comments and guidance. For ease of reviewing, all the significant revisions in the revised manuscript are highlighted in yellow color.

Review #1

Point 1: Before the manuscript is accepted, I'd like to see the authors add 95%CI to their tables so that future researchers may have an opportunity to interpret the analysis as well.

Response 1: Thank you for your helpful comment.

We are sorry for the mistake in our study. Per your advice, we have added 95%CI to our tables in the revised manuscript. Please see the tables and supplementary tables in the revised manuscript.

Thanks!

We appreciate all your time and effort in helping us improve and clarify our manuscript. We hope our revision has satisfactorily addressed the comments and will be acceptable.

Sincerely,

Feng Zhang, PhD.

Tel: 86-29-82655091. Fax: 86-29-82655032.
